# A survey of gyrodactylid parasites on the fins of *Homatula variegata* in central China

Xiaoning Chen[1,2], Biao Wang[1], Jianzhen Nie[1], Ping You[1]*

**1** College of Life Sciences, Shaanxi Normal University, Xi'an, Shaanxi, China, **2** Shaanxi Key Laboratory for Animal Conservation, Shaanxi Institute of Zoology, Xi'an, Shaanxi, China

* youping@snnu.edu.cn

**Data Availability Statement:** All relevant data are within the manuscript and its Supporting Information files. The raw data have been uploaded into Dryad under the following https://doi.org/10.5061/dryad.5x69p8d0p.

## Abstract

In this study, two parasites on the fins of *Homatula variegata* were recorded from March to September 2016. A dissection mirror was used to examine the distribution and quantity of the ectoparasitic *Gyrodactylus* sp. and *Paragyrodactylus variegatus* on the host *Homatula variegata* in different seasons. The present study explored possible explanations for the site specificity of gyrodactylid parasites in 442 *Homatula variegata* infected with 4307 *Gyrodactylus* sp. (species identification is incomplete, only characterized to the genus level) and 1712 *Paragyrodactylus variegatus*. These two gyrodactylid parasites were collected from fish fins, and the fish were harvested in China's Qinling Mountains. The results indicated that the highest number of *Gyrodactylus* sp., which was numerically the dominant species, appeared on the fish fins in April, while the highest number of *Paragyrodactylus variegatus* was found on the fish fins in March. The two parasite species appeared to be partitioned spatially, with *Gyrodactylus* sp. occurring more frequently on pectoral and pelvic fins, and *P. variegatus* occurring more frequently on caudal fins. However, *Gyrodactylus* sp. appeared to occur on fish of all lengths, while *P. variegatus* tended to occur more abundantly on shorter fish rather than on longer fish. At lower *Gyrodactylus* sp. infection levels (<100), the pelvic and pectoral fins were the main locations of attachment, followed by the dorsal fin. For infections of more than 100 parasites, more samples of *Gyrodactylus* sp. were located on the pectoral fin. For a low number of *Paragyrodactylus variegatus* infections (<100), the pelvic and pectoral fins were the preferred locations of attachment, followed by the caudal fin. Between April and September, there were many monogenean parasites on fish fins, and the fish size was within the range of 5–10 cm. However, when a fish was longer than 10 cm long, the number of parasites on its fins greatly decreased.

## Introduction

Among the members of the class Monogenea, viviparous gyrodactylids are one of the most common parasites in wild and cultured fish, causing great ecological and economic harm [1]. Some gyrodactylids show significant microhabitat specificity, but this is highly variable among species. Some researchers focused on the site preference of *Gyrodactylus turnbulli* on *Poecilia*

**Funding:** This study was supported by grants from the National Natural Science Foundation of China (31872203) and the Natural Science Foundation of Shaanxi Province (2017JM3014). The funders had no role in study design, data collection and analysis, decision to publish, or preparation of the manuscript.

**Competing interests:** The authors have declared that no competing interests exist.

*reticulata* (guppy) in an experimental environment. Studies have found that lymphocytes in fish epithelial tissues have a direct effect on parasites after they come into contact with a host. The host's innate and adaptive immune system determines where the parasite lives [2–4].

However, the distribution of parasites on fish is strongly correlated with the age of infection. Water quality and water nutrition are factors that determine the abundance of fish parasites. However, changes in water temperature and seasons are also determinants of parasite abundance [5–6]. Other groups have reached similar conclusions by studying the behavior of parasites (*G. colemanensis*) on *Salvelinus fontinalis* fry [7]. Parasites attached to any part of fish epithelial tissue. In particular, many parasites occur on the edges of the tail, pectoral and peritoneal fins. Parasites periodically migrate to the edges of the fins and can travel through the body to reach other fins [8]. Recently, the fish *Homatula variegata* (Dabry de Thiersant, 1874) has become an increasing concern due to its potential aquaculture in China [9]. *Gyrodactylus* sp. and *Paragyrodactylus variegatus* (You, King, Ye and Cone, 2014) [10] are two parasites that are found on the fins and, occasionally, the body surface, of *H. variegata* in Xunyangba. However, gyrodactylids that live on the surface of fish appear to be less specific in terms of their environmental requirements and therefore occur in a variety of locations. This assumption, has led to a lack of information on the positioning of parasites on this fish; most authors locate parasites on the major branches of the body of a fish, such as the gills or torso [11]. However, no specific studies on the distribution of gyrodactylid parasites in *Homatula variegata* have been performed. Therefore, this study attempts to describe the position specificity of *Gyrodactylus* sp. in *Homatula variegata*.

## Materials and methods

### Ethical note

This study was approved by the Animal Care and Use Committee of Shaanxi Normal University.

### Study area and sample collection

The fish (*Homatula variegata*) were collected (n = 442) with seine nets from late March to late September 2016 in Xunyangba (33.33˚N, 108.33˚E), Ningshan, County located on the southern slopes of the Qinling Mountains in Shaanxi Province, central China. The water temperatures on the collection days were recorded (Table 1).

Each fish was individually placed in a plastic tank filled with filtered river water and was transported to a field laboratory and examined within one hour. The fish were euthanized with excessive eugenol anesthetic fluid and fixed with 5–10% formalin. The total length of each fish was recorded, and the fins were examined for the presence of parasites that were removed and immediately identified on temporary wet mounts.

**Table 1. Morphometrics of the collected *Homatula variegata*.**

| Month | Water temperature (˚C) | Number of Fish | Body length (cm) | Average body length mean± SE (cm) |
|---|---|---|---|---|
| March | 7 | 43 | 3.60–13.60 | 5.70±0.372 |
| April | 13 | 40 | 3.80–13.80 | 8.43±0.425 |
| May | 17.5 | 42 | 4.10–11.70 | 7.15±0.303 |
| June | 21 | 46 | 4.50–14.60 | 8.37±0.370 |
| July | 23 | 85 | 4.50–14.00 | 9.10±0.211 |
| August | 17 | 100 | 3.10–13.90 | 9.36±0.218 |
| September | 15 | 86 | 3.40–13.80 | 7.91±0.287 |

These two species of *Gyrodactylus* were found under a dissecting microscope (OLYMPUS, SZ61, 45X), Then they were placed on glass slides that had drops of glycerin-water with pointed ophthalmic forceps. If there was a cap-like bone piece structure covering the base of the central hook, then it was recorded as *Paragyrodactylus variegatus*; otherwise, it was recorded as *Gyrodactylus* sp.

The parasites were stored in formalin. Almost all the parasites stuck to the skin after immobilization. Voucher specimens of the parasites and host were deposited in the Fish Disease Laboratory, Shaanxi Normal University (Accession number: *H. variegata*: Acc.HV20160012; *Gyrodactylus* sp.: Acc.GS20160001 and *P. variegatus*: Acc.PV20160001).

## Analysis of parasite location

The different numbers of parasites on each fin corresponding to different fish body lengths were examined, from March to September, 2016. The parasite species and location of parasites on the host's fins were examined by using a two-way ANOVA, with the number of parasites on different fins used as the dependent variable. The microhabitat occurrence for each gyrodactylid species was determined by observing its position on the fins. The distributions of each gyrodactylid species on each of the different fins were compared by two-way ANOVA with multiple comparisons (Tukey's HSD test) to assess the significance of the difference. The significance level was set at $p < 0.05$.

## Relationships between water temperature, parasite load location and fish size

To test for the overall effects of water temperature and fish body length on the distribution of the number of parasites on the host fins, a generalized linear model (GLM) was also built using water temperature or fish body length as predictors. To further explore the relationships of fish size (length) and water temperature with the number of parasites on different fins, Spearman correlation statistical analysis was conducted using SPSS (Statistical Package for Social Sciences, v21).

## Results

### The number dynamics of gyrodactylid parasites and distribution of parasites on fish fins

The fish (*Homatula variegata*) were collected (n = 442) with seine nets from late March to late September 2016 in Xunyangba (33.33°N, 108.33°E), Ningshan County located on the southern aspects of the Qinling Mountains in Shaanxi Province. The water temperatures on the collection days were recorded. We measured the body length of each host. We specifically collected two gyrodactylid parasites from the host fins. The species and number of parasites on different fins were recorded in detail. Tukey's HSD test was used to analyze the significance of parasite distribution on the fins, and GLM was used to analyze the effect of host length and water temperature on the location of parasites. To explore the specific effect of water temperature and length on the number of parasites on fins, a Spearman correlation analysis was performed.

Temporal changes in the number of parasites on the fins were analyzed. The highest number of *Gyrodactylus* sp. on the fish fins appeared in April, while the highest number of *P. variegatus* on the fish fins occurred in March. The number of *Gyrodactylus* sp. and *P. variegatus* showed roughly similar trends, with the number of the two parasites (mean ± SE) on fins decreasing in May (4.17 ± 0.74 and 2.6 ± 0.43, respectively); the number of parasites on the fins decreased in August (2.09 ± 0.31 and 2.17 ± 0.27) and increased in September (5.53 ± 0.43

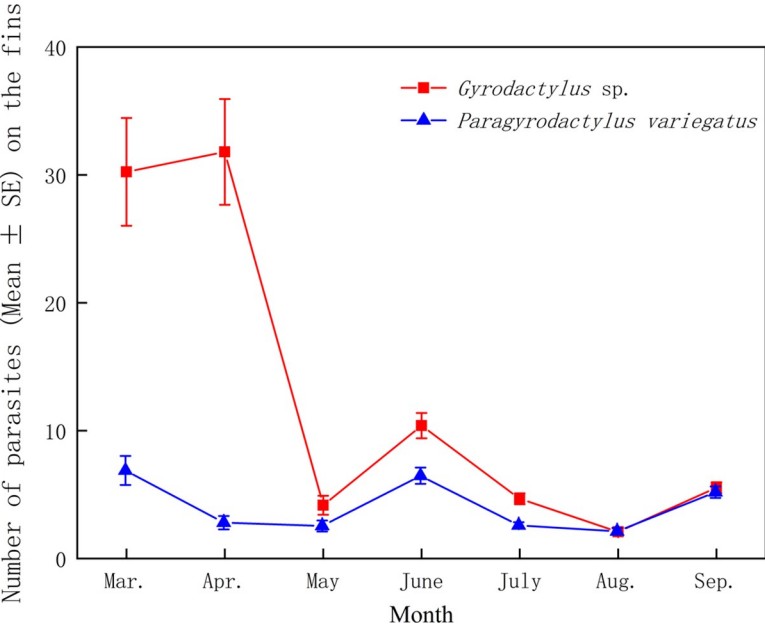

**Fig 1. Parasites number (mean±SE) of *Gyrodactylus* sp. and *Paragyrodactylus variegatus* on total fish fins by month.**

and 5.23 ± 0.45) (Fig 1). Of the 7 months, the number of the two gyrodactylid parasites on the fins were lowest in August. There was no significant correlation between number of parasites and the water temperature, which ranged from 7˚C (in March) to 23˚C (in July) (*Gyrodactylus* sp.: $r = -0.149$, $p = 0.751$; *P. variegatus*: $r = 0.090$, $p = 0.847$).

On the other hand, the number of *P. variegatus* on the pectoral fin was relatively high in May and July. The number of parasites detected on the dorsal fins and anal fins was relatively low (Fig 2A). The number of parasites on different fins decreased significantly during April and May. In June, the number of *Gyrodactylus* sp. increased on the pectoral fin (Fig 2B).

For *Gyrodactylus* sp., there was a significant difference (two-way ANOVA, $F = 11.97$, $df = 4$, $p < 0.001$) among the mean number (mean ± SE) of parasites that were distributed on the five fins. All data were examined from a total of 442 specimens of *Homatula variegata* from Xunyangbain the Qinling Mountains of Shaanxi Province, central China, which were collected from March to September 2016. The total length of the host (± 0.1 cm) ranged from 3.1 to 14.6 cm. The number of fish and the water temperature for each sampling period are recorded in Table 1. For *P. variegatus*, there was also a significant difference (two-way ANOVA, $F = 30.94$, $df = 4$, $p < 0.001$) among the mean numbers (mean ± SE) of parasites that were distributed on the five fins. In general, the mean number (mean ± SE) of *Gyrodactylus* sp. infecting different fin parts was higher than that of *P. variegatus* (Fig 2) by month. However, we also detected a contrasting pattern between the densities of specific parasitic infections of the fins. Although there was no difference between the pectoral fins and the pelvic fins of the two gyrodactylid parasites (Tukey's HSD, $df = 4$, $p > 0.05$), and the number of parasites on these two fins was higher than that on any other fin (Tukey's HSD, $df = 4$, all $p < 0.001$; Fig 3). The patterns of parasite number were comparable in *Gyrodactylus* sp. and *P. variegatus* but were clearly different from those on respective fins (two-way ANOVA, estimated marginal mean test).

By comparison, the number of *Gyrodactylus* sp. was significantly higher on pelvic fins and pectoral fins than on other fins (test of between-subjects effects, F (4, 4410), $p < 0.001$). Both

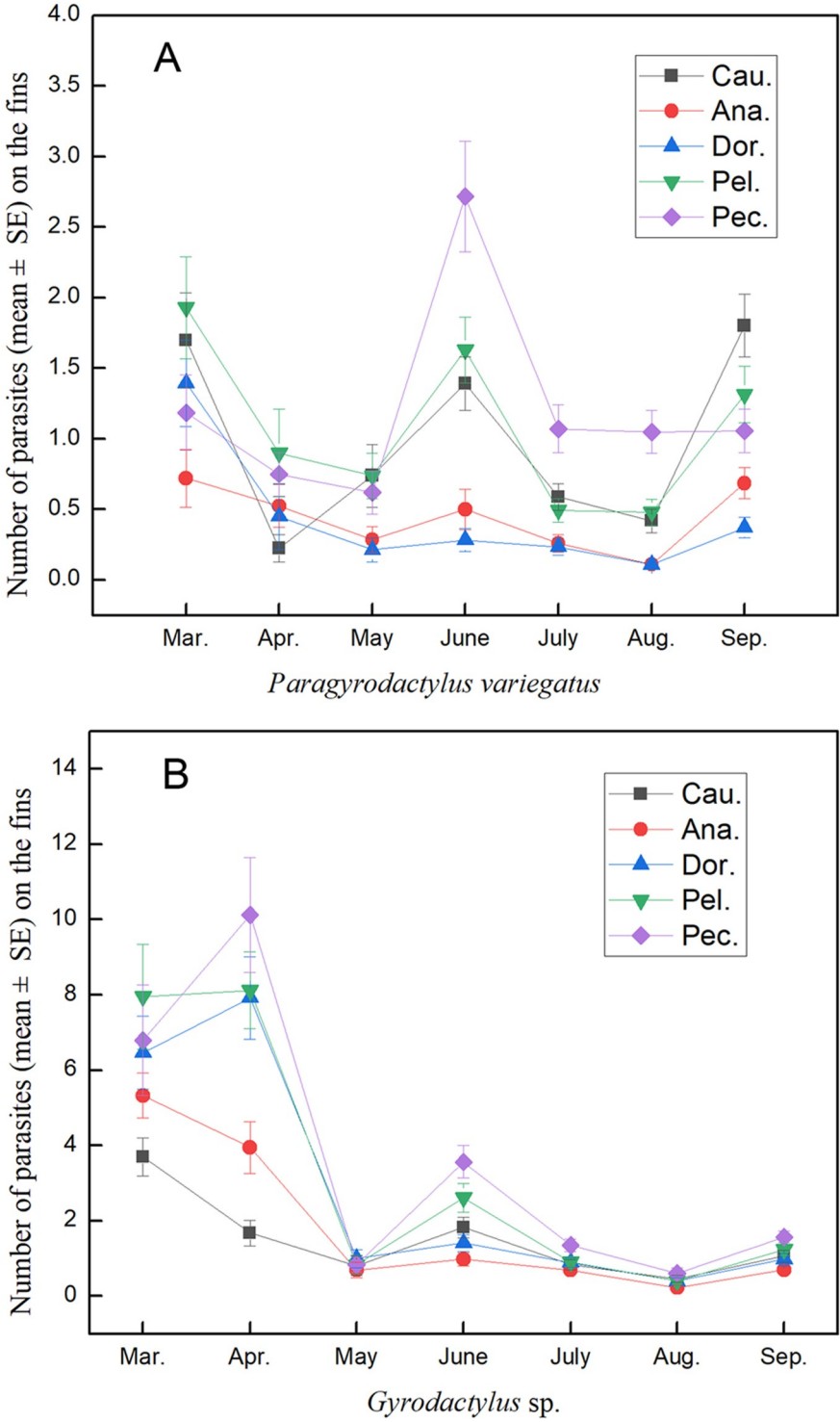

**Fig 2. Parasites number (mean±SE) on different fish fins monthly.** A. *Paragyrodactylus variegatus*, B. *Gyrodactylus* sp.; the abbreviations Ana., Cau., Dor., Pec., and Pel., indicate, anal fin, caudal fin, dorsal fin, pectoral fin and pelvic fin, respectively.

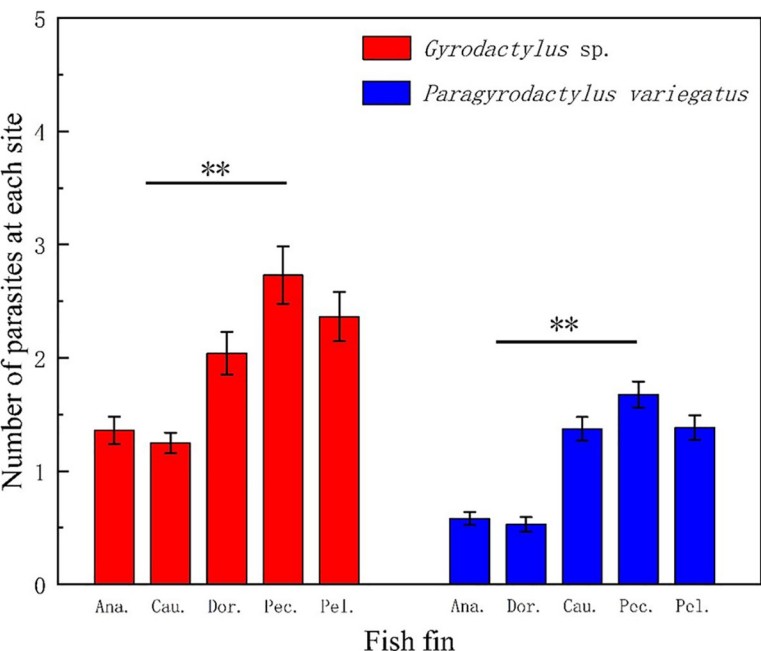

**Fig 3. Number of *Gyrodactylus* sp. and *Paragyrodactylus variegatus* on different fish fins, the abbreviations Ana., Cau., Dor., Pec., and Pel., indicate, anal fin, caudal fin, dorsal fin, pectoral fin and pelvic fin, respectively.** $^{*}p < 0.05$; $^{**}p < 0.01$.

parasite species appeared in moderate amounts on the dorsal fins (*Gyrodactylus* sp.) and caudal fins (*P. variegatus*), respectively (Fig 3). On the other hand, the number of *P. variegatus* was also significantly higher on pelvic, pectoral and caudal fins than that on anal and dorsal fins (Tukey's HSD, *df* = 4, both $p < 0.001$). No other significant differences in the number of specific parasitic infections among host fins were detected. At lower levels of infection, parasites preferentially colonized the pelvic, pectoral and dorsal fins (Table 2). Most fins were infected with a small number of parasites. The number of *P. variegatus* present in fish was less than 100. As the number of parasites on each fish fin increased by 11 to 100, pelvic fins continued to be the main area of attachment, followed by the pectoral fin. However, at a level of 100 or more parasites per fish, relatively few *Gyrodactylus* sp. were observed on some fins, such as caudal fins. The total number of *Gyrodactylus* sp. was higher than that of *P. variegatus* on all fins per fish. The cause of this phenomenon needs further study.

**Table 2. The site specificity of *Gyrodactylus* sp. and *Paragyrodactylus variegatus* on four different range of infection of *Homatula variegata* (the average number of infections is on each fin).**

| Gyrodactylus sp. | | | | | | | Paragyrodactylus variegatus | | | | | | |
|---|---|---|---|---|---|---|---|---|---|---|---|---|---|
| Nop | Nof | Cau | Ana | Dor | Pel | Pec | Nop | Nof | Cau | Ana | Dor | Pel | Pec |
| 0 | 53 | 0 | 0 | 0 | 0 | 0 | 0 | 79 | 0 | 0 | 0 | 0 | 0 |
| 1~10 | 271 | 0.727 | 0.539 | 0.738 | 0.86 | 1.251 | 1~10 | 331 | 0.925 | 0.378 | 0.347 | 0.876 | 1.187 |
| 11~100 | 114 | 3.07 | 3.693 | 5.579 | 6.342 | 6.781 | 11~100 | 35 | 3.457 | 1.571 | 1.657 | 4.057 | 3.657 |
| 101~200 | 3 | 3 | 16 | 29 | 36.667 | 41.667 | 101~200 | 0 | N/A | N/A | N/A | N/A | N/A |

Nop: Range of parasites number; Nof: number of fish; Cau: caudal fin; Ana: anal fin; Dor: dorsal fin; Pel: pelvic fin; Pec: pectoral fin.

**Table 3. Parasitism by *Gyrodactylus* sp. and *Paragyrodactylus variegatus* on *Homatula variegata*.**

| *Gyrodactylus* sp. | | | | | | | | *Paragyrodactylus variegatus* | | | | | | | |
|---|---|---|---|---|---|---|---|---|---|---|---|---|---|---|---|
| Month | Fs (cm) | Nof | Cau | Ana | Dor | Pel | Pec | Month | Fs (cm) | Nof | Cau | Ana | Dor | Pel | Pec |
| March 2016 | 1–5 | 25 | 85 | 97 | 125 | 134 | 68 | March 2016 | 1–5 | 25 | 22 | 6 | 20 | 20 | 12 |
|  | 5–10 | 14 | 68 | 104 | 102 | 118 | 134 |  | 5–10 | 14 | 31 | 19 | 28 | 47 | 23 |
|  | 10–15 | 4 | 6 | 28 | 51 | 90 | 90 |  | 10–15 | 4 | 20 | 6 | 12 | 16 | 16 |
| April 2016 | 1–5 | 7 | 9 | 16 | 31 | 32 | 29 | April 2016 | 1–5 | 7 | 0 | 0 | 0 | 2 | 2 |
|  | 5–10 | 19 | 48 | 71 | 145 | 168 | 192 |  | 5–10 | 19 | 7 | 9 | 13 | 12 | 13 |
|  | 10–15 | 14 | 10 | 71 | 141 | 125 | 184 |  | 10–15 | 14 | 2 | 12 | 5 | 22 | 15 |
| May 2016 | 1–5 | 8 | 9 | 12 | 19 | 15 | 12 | May 2016 | 1–5 | 8 | 2 | 3 | 1 | 4 | 6 |
|  | 5–10 | 32 | 23 | 17 | 22 | 20 | 23 |  | 5–10 | 32 | 29 | 9 | 8 | 26 | 22 |
|  | 10–15 | 3 | 2 | 0 | 1 | 1 | 1 |  | 10–15 | 3 | 0 | 0 | 0 | 1 | 0 |
| June 2016 | 1–5 | 6 | 8 | 2 | 9 | 3 | 29 | June 2016 | 1–5 | 6 | 11 | 1 | 2 | 1 | 13 |
|  | 5–10 | 31 | 67 | 40 | 45 | 77 | 116 |  | 5–10 | 31 | 46 | 20 | 9 | 54 | 98 |
|  | 10–15 | 10 | 9 | 3 | 11 | 15 | 21 |  | 10–15 | 10 | 7 | 2 | 2 | 11 | 15 |
| July 2016 | 1–5 | 1 | 0 | 0 | 1 | 1 | 0 | July 2016 | 1–5 | 1 | 2 | 3 | 1 | 0 | 0 |
|  | 5–10 | 57 | 52 | 40 | 52 | 52 | 85 |  | 5–10 | 57 | 31 | 14 | 14 | 25 | 67 |
|  | 10–15 | 27 | 19 | 18 | 23 | 24 | 30 |  | 10–15 | 27 | 16 | 7 | 5 | 17 | 23 |
| Aug. 2016 | 1–5 | 4 | 1 | 0 | 0 | 0 | 0 | Aug. 2016 | 1–5 | 4 | 0 | 0 | 0 | 0 | 3 |
|  | 5–10 | 57 | 29 | 15 | 19 | 31 | 52 |  | 5–10 | 57 | 27 | 6 | 8 | 35 | 82 |
|  | 10–15 | 39 | 16 | 7 | 21 | 9 | 8 |  | 10–15 | 39 | 15 | 5 | 3 | 13 | 20 |
| Sep. 2016 | 1–5 | 19 | 27 | 11 | 13 | 24 | 32 | Sep. 2016 | 1–5 | 19 | 60 | 11 | 6 | 17 | 8 |
|  | 5–10 | 47 | 54 | 45 | 42 | 66 | 86 |  | 5–10 | 47 | 84 | 34 | 19 | 77 | 51 |
|  | 10–15 | 20 | 10 | 4 | 29 | 16 | 17 |  | 10–15 | 20 | 11 | 14 | 3 | 7 | 16 |

Fs: Range of fish body size; Nof: number of fish; Cau: caudal fin; Ana: anal fin; Dor: dorsal fin; Pel: pelvic fin; Pecf: pectoral fin.

## Relationships between the number of parasites on the fins and the body length of the fish and water temperature

Of the 442 fish collected, 135 were less than 7 cm in length, accounting for 30.5% of the total; 183 fish had a body length of more than 7 cm but less than 10 cm, accounting for 41.4% of the total;124 fish had a body length of more than 10 cm, accounting for 28.1% of the total. Interestingly, throughout the study period (March to September 2016), host fish were infected with the two gyrodactylid parasites at significantly fluctuating levels (Table 3). By examining the fins of all the fish samples, a large number of the two parasites were observed on the fins of the fish hosts with relatively shorter body lengths (1–5 cm). There were fewer of these two parasites on the fins of fish hosts with relatively larger body lengths (greater than 5 cm).

GLM analysis showed that host body length had a significant effect on the number of *Gyrodactylus* sp. on the pectoral fins. Host body length had a significant effect on the number of *P. variegatus* on the caudal and anal fins. Water temperature had a significant effect on the number of these two gyrodactylid parasites on all fins, and the specific differences need further analysis (Table 4).

After statistical analysis, water temperature and host body length had a high negative correlation (spearman) with the number of *Gyrodactylus* sp. on the fins (Table 5). However, these factors were relatively less relevant to the number of *P. variegatus* on the fins. Independently, the correlation between water temperature and the number of *Gyrodactylus* sp. on the fins was higher. Interestingly, the water temperature was negatively related to not only the number of *Gyrodactylus* sp. on the fins but also the host body length but to a lesser extent. These results are essentially consistent with the HSD analysis and GLM analysis (Table 4).

**Table 4. Results of the generalized linear models (GLM) used to determine the body length and water temperature on the total gyrodactylid parasite number on the different fins of *Homatula variegata*.**

| Dependent variable [a] | Degrees of freedom [b] | Coefficient | p value |
|---|---|---|---|
| Total fins (*Gyrodactylus* sp.) | 105 | Intercept | <0.001 |
| | | Host body size(length) | 0.006* |
| | | water temperature | <0.001** |
| Cau. fins (*Gyrodactylus* sp.) | 82 | Intercept | <0.001 |
| | | Host body size(length) | 0.692[Δ] |
| | | water temperature | <0.001** |
| Ana. fins (*Gyrodactylus* sp.) | 85 | Intercept | <0.001** |
| | | Host body size(length) | 0.101[Δ] |
| | | water temperature | <0.001** |
| Dor. fins (*Gyrodactylus* sp.) | 92 | Intercept | <0.001** |
| | | Host body size(length) | 0.130[Δ] |
| | | water temperature | <0.001** |
| Pel. fins (*Gyrodactylus* sp.) | 95 | Intercept | <0.001** |
| | | Host body size(length) | 0.066[Δ] |
| | | water temperature | <0.001** |
| Pec. fins (*Gyrodactylus* sp.) | 93 | Intercept | <0.001** |
| | | Host body size(length) | <0.001** |
| | | water temperature | <0.001** |
| Total fins (*Paragyrodactylus variegatus*) | 98 | Intercept | <0.001** |
| | | Host body size(length) | <0.001** |
| | | water temperature | <0.001** |
| Cau. fins (*Paragyrodactylus variegatus*) | 75 | Intercept | <0.001** |
| | | Host body size(length) | 0.005* |
| | | water temperature | 0.003* |
| Ana. fins (*Paragyrodactylus variegatus*) | 62 | Intercept | <0.001** |
| | | Host body size(length) | 0.032* |
| | | water temperature | 0.038* |
| Dor. fins (*Paragyrodactylus variegatus*) | 58 | Intercept | <0.001** |
| | | Host body size(length) | 0.139[Δ] |
| | | water temperature | 0.006* |
| Pel. fins (*Paragyrodactylus variegatus*) | 80 | Intercept | <0.001** |
| | | Host body size(length) | 0.085[Δ] |
| | | water temperature | <0.001** |
| Pec. fins (*Paragyrodactylus variegatus*) | 90 | Intercept | <0.001** |
| | | Host body size(length) | 0.357[Δ] |
| | | water temperature | 0.001* |

a. The number of parasites in the corresponding location

b. Sig. of *Omnibus* test

Δ. No Significance ($p > 0.05$)

* Significance ($p < 0.05$)

** Extremely Significance ($p < 0.001$)

## Discussion

Although it has been reported in China, little research has been conducted on the survival of gyrodactylids on *Homatula variegata*, for which a total of two genera were found: *Gyrodactylus* (von Nordmann, 1832) and *Paragyrodactylus* (Gvosdev et Martechov, 1953). Our investigation

**Table 5. Spearman coefficients between body length of fish and water temperature with gyrodactylid parasites on the fins.**

| Parasite | Water temperature *vs* Number of parasites on fins. *Spearman correlation coefficient and significance.* | Host body length *vs* Number of parasites on fins. *Spearman correlation coefficient and significance.* |
|---|---|---|
| *Gyrodactylus* sp. | $r$ = -0.351, $p$ < 0.001 | $r$ = -0.269, $p$ < 0.001 |
| *Paragyrodactylus variegatus* | $r$ = -0.183, $p$ < 0.001 | $r$ = -0.197, $p$ < 0.001 |

Host samples = 442

revealed that *Gyrodactylus* sp. and *P. variegatus* survive on this host. The microhabitat of monogeneans living on fins has been investigated by many authors [12–14]. Monogeneans exhibit the characteristics of aggregate parasitism. For example, benedeniines are significantly parasitic on specific fins [12]. Studies have found that parasites that are attached to the dorsal fins or pelvic fins of fish may be designed to evade host predation, competition, and local immune responses [15, 16]. In addition, each developmental cohort that inhabits different fish fins can receive exclusive food and spatial resources [14].

In the present study, the two species of parasites appear to have subtle spatial partitions in their common resources. *Gyrodactylus* sp. occurred most frequently on the pectoral and pelvic fins, while *P. variegatus* occurred on the caudal fins.

In this study, we investigated the average water temperature of the sampling points during sampling. Water temperature is thought to be a factor affecting a parasite's ability to reproduce [17–19]. Moreover, there is a certain degree of correlation between the water temperature and number of parasites on fins, but the influence of water temperature is distinct in different parasite species [20]. The relationship between temperature and parasite reproduction is complex. Some literature has noted that the number of parasites increases with increasing water temperature [21, 22]. On the other hand, for some species, elevated temperatures can be a limiting factor for survival and reproduction [23, 24]. In our study, we found that the number of *Gyrodactylus* sp. and *P. variegatus* on the fins of *H. variegata* showed a different trend; specifically, the number of *Gyrodactylus* sp. number on the fins reached its highest point in April, gradually declined in summer and increased again in autumn (Fig 1). Some previous findings support the results of increased numbers in summer [18, 25], which is consistent with our findings. In addition, although the water temperature in July (23˚C) was higher than that in June (21˚C), the volumes of the two fin parasites in July were lower than those in June. The reason may be that the immunity of the host fish increases with increasing water temperature, thus leading to a decline in the number of parasites. Previous studies have demonstrated the same results in higher levels of infection and with weaker host immunity [20, 26]. Another reason for this outcome may have been the changes in aquatic environmental factors that are were caused by the increased water temperatures in July, which led to a decrease in the number of aquatic environmental factors. In addition to temperature, photoperiod, salinity and water flow can influence the success of infection. Studies have found that host fish are only infected by monogenean parasites during the day, and low-temperature and high-salt waters are more conducive to parasitic infections of fish [27]. Interestingly, the number of *Gyrodactylus* sp. on the fish fins decreased significantly relative to that of *P. variegatus*, in May. The cause for this is unknown at this time but could involve interspecific competition. The number of *Gyrodactylus* sp. rose again in September, possibly due to changes in the water flow rate. More detailed work on this topic is clearly required. We found a negative correlation between the number of *Gyrodactylus* sp. and the body length of *H. variegata*, and there was a relatively weak negative correlation between the number of *P. variegatus* and the body length of the fish. This finding is

similar to the results of some previous studies. Thus, there was a negative correlation between parasite species richness and fish body size [28–30]. Due to the increase in parasite amount, in comparison to larger fish, smaller fish hosts may be more susceptible to disease [1]. However, some previous literature observed the opposite, in which there was a positive correlation between the number of parasites on the fins and the size of the host [31–34]. Some researchers have noted that the relationship between the number of parasites on the infected sites of fish and the length of the host should also be highlighted. This effect is more pronounced in small fish, which have a higher number of parasites on the body surface [35, 36]. Another possible reason might be that fish with a longer body length may be found in microhabitats with less exposure to parasitic infections [30]. Fish use group behavior and immune responses to reduce the risk of parasites [37]. Perhaps large fish are better suited to finding groups. The current scope of the study reveals that there was less aggregation of parasites on large *Homatula variegata* fins, which is consistent with the idea that the chances of avoiding infection are enhanced. It is important to emphasize, however, that not only the host size but also the ecology of each host species affects the species richness of the parasite [34].

Different kinds of *Gyrodactylus* are parasitic to different parts of the host. *Gyrodactylus masu* is found on the body surface of salmonids, and the fins, gill arches and gill filaments are the main locations [38]. By observing the parasitic behavior of the five species of *Gyrodactylus* parasites, researchers found that four of the parasites prefer to parasitize the fish surface and fish gills. When studying two parasites (*G. colemanensis* and *G. salmonis*) on the surface of the salmonids, most of the *G. colemanensis* were attached to the edge of the fin. *Gyrodactylus salmonis* attached to the head and body surface of the fish [7]. Studies have found that different types of *Gyrodactylus* have different haptor shapes, which may lead to differences in their habitats [7, 39]. The morphology of the haptors of each *Gyrodactylus* species is probably adapted to the surface of their host.

## Conclusions

In summary, through the investigation of two parasites that infect *Homatula variegata*, we found that (1) the highest number of *Gyrodactylus* sp. on the fins appeared in April and March, whereas the number of *Paragyrodactylus variegatus* on the fins appeared in June. That is, the peak number of the two parasites on the fins showed a time niche separation. However, the trend in the number of parasites on the fins was similar during from May to September. The number of the two parasites on the fins of the host rise again in the autumn. (2) The two gyrodactylid parasites seemed to partition their common resources spatially. *Gyrodactylus* sp. preferred to parasitize the pectoral and pelvic fins, while *P. variegatus* preferred to parasitize the caudal fins. This may be explained by the avoidance of predation competition and the host's local immune response. This selection mode can be used as a potential delivery policy in gyrodactylids. The main factors leading to the preference for specific habitats have not yet been determined and may be linked to physiological, environmental, ecological and physical factors. More research is required to clarify this preference.

## Supporting information

**S1 Data.**
(XLSX)

## Acknowledgments

The authors sincerely thank Fenhong Li, Fei Ye, Jun Yan, Junqing Jia and Xue Lan (College of Life Sciences, Shaanxi Normal University) for collecting experimental samples and data, and Dr. David K. Cone (Department of Biology, Saint Mary's University, Canada) and Fei Ye for their valuable suggestions for the manuscript.

## Author Contributions

**Data curation:** Xiaoning Chen, Biao Wang, Jianzhen Nie, Ping You.

**Formal analysis:** Xiaoning Chen, Jianzhen Nie.

**Funding acquisition:** Ping You.

**Investigation:** Xiaoning Chen, Biao Wang, Jianzhen Nie.

**Project administration:** Ping You.

**Resources:** Xiaoning Chen, Biao Wang.

**Software:** Xiaoning Chen.

**Writing – original draft:** Xiaoning Chen.

**Writing – review & editing:** Ping You.

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
