## [Decision Letter · Decision Letter 0]

17 Jan 2020

PONE-D-19-26341

A survey of gyrodactylid parasites on the fins of Homatula variegata in central China

PLOS ONE

Dear Dr. You,

Thank you for submitting your manuscript to PLOS ONE. After careful consideration, we feel that it has merit but does not fully meet PLOS ONE’s publication criteria as it currently stands. Therefore, we invite you to submit a revised version of the manuscript that addresses the points raised during the review process.

Both reviewers agree that you manuscript contains valuable information, however, there are several shortcomings related to the lack of species identification for one of the parasite species, as well as to data acquisition and analysis. Additionally, you should have your manuscript edited by a scientist who is fluent in the English language.

We would appreciate receiving your revised manuscript by Mar 02 2020 11:59PM. To enhance the reproducibility of your results, we recommend that if applicable you deposit your laboratory protocols in protocols.io, where a protocol can be assigned its own identifier (DOI) such that it can be cited independently in the future. For instructions see: http://journals.plos.org/plosone/s/submission-guidelines#loc-laboratory-protocols

We look forward to receiving your revised manuscript.

Kind regards,

Ulrike Gertrud Munderloh, Ph.D.

Academic Editor

PLOS ONE

Journal Requirements:

Reviewers' comments:

Reviewer's Responses to Questions

**Comments to the Author**

1. Is the manuscript technically sound, and do the data support the conclusions?

Reviewer #1: Yes

Reviewer #2: No

2. Has the statistical analysis been performed appropriately and rigorously? 

Reviewer #1: Yes

Reviewer #2: No

3. Have the authors made all data underlying the findings in their manuscript fully available?

Reviewer #1: Yes

Reviewer #2: Yes

4. Is the manuscript presented in an intelligible fashion and written in standard English?

Reviewer #1: Yes

Reviewer #2: No

5. Review Comments to the Author

Reviewer #1: The authors did a great work in the manuscript and the analysis of the specimens collected. But I consider that this paper should be submitted to another journal rather than PlosOne. Although, the data and the species reported are something new for China, I would identify the Gyrodactylus species to give a better approach to the study, it is been demostrated that a single fish species can be infected by different gyrodatylid species and this can give different results, and the numbers in this study, for example can be mistaken. The infection dynamics of different species are very different, and this can be mistaken (in the order to imply, that is only one species that they found in the present study).

Abstract:

The authors mentioned that they collected the Gyrodactylus specimens with fishing nets and scorpions. I think they collected the fish with these nets, rather than the parasites. Please change.

Introduction:

The geograhical locality is mentioned with the coordinates, please delete this, that should be mentioned in the Material and Methods (M&M) subject. Delete it and move it to this section.

Materials and Methods:

The authors did not mentioned the number of fish they sampled (this is mentioned in Abstract, but not in M&M, please do so.

The accession numbers should be written as follows: H. variegata Acc. No. HV20160012 or Accession numbers H. variegata HV20160012, but is better the first one.

The animal processing must go with the name of the institution, number of permission, city and country.

Results:

Please check the manuscript for comments on this part.

How can the authors be sure that they only have 1 species of Gyrodactylus infecting the host?

Discussion:

Please check the manuscript for comments on this part.

Conclusions:

The authors have some works wrongly spelled. Please check and change.

Acknowledgements:

The research was funded by the National Natural Science Foundation of China (Project number 31872203) and the Natural Science Foundation of Shaanxi Province (Project number 2017JM3014).

References:

Most of the references are miss spelled and don’t have a unified style. Please change it in the correct journal form.

Tables:

Table 2. I think that if the authors have shown the data in table1 per month, they should do the same here with the parasite data. They must add the abbreviations here of the different fins names rather than in Table 3.

Figures:

Figure 2. The names on axis X there is a missing space between parasites the brackets and on, please change it.

Figure 5. I don’t see the point of adding this figure, it doesn’t show anything.

Reviewer #2: The ms covers a potentially important topic well worth publication, but as presented, has several important shortcomings. Even though original, unpublished experimental data are presented, the methods used to generate and interpret the results are insufficient to replicate the work and use the data to their full potential. Additionally, inconsistent and not widely accepted parameters, not fully described in the materials and methods are used throughout the text. Conclusions are not really well supported by the data as presented, and are nevertheless used as a base for unnecessary, non-related speculation that could be omitted.

6. PLOS authors have the option to publish the peer review history of their article (what does this mean?). If published, this will include your full peer review and any attached files.

Reviewer #1: No

Reviewer #2: No

---

## [Author Response · Author response to Decision Letter 0]

26 Feb 2020

Dear Dr. Ulrike Gertrud Munderloh,

Thank you for your letter and for the reviewers’ comments concerning our manuscript entitled ‘A survey of gyrodactylid parasites on the fins of Homatula variegata in central China’ (PONE-D-19-26341). The comments were very helpful for revising our paper. We have studied the comments carefully and have made revisions accordingly, which we hope meet with approval. Revisions have been made using the Track Changes feature in MS Word. The revisions to the paper are described in our responses to the academic editor and reviewers’ comments below.

Thank you in advance.

Cheers,

Ping You

Notes on content modification

Thank you very much for your noble opinions and suggestions. According to the requirements, we have revised the full manuscript peer-to-peer. The main content is as follows

Abstract

Question：

Not clear what this is; please specify, “scorpions”

Answer：

It is a pair of tweezers used to clamp the parasite from the fish's body. Revised the description of the sentence.

Question：

Please revise/rephrase: if the species identity was not established, and parasites were only characterized to the genus level, this should be reflected in the text.

Answer：

The description of the species name has been revised. 

The present study will explore possible explanations for the site specificity of gyrodactylidae parasites in 442 Homatula variegata infected with 4307 Gyrodactylus sp. (Species identification is incomplete, only characterized to the genus level) and 1712 Paragyrodactylus variegatus.

Question：

A bit confusing, if both taxa occur more frequently on the same two areas...

Answer：

Through GLM analysis, it was further found that the fin positions of the two parasites were not exactly the same. Revised the description of the sentence

Question：

Are you sure that you use fish nets to collect the parasites, rather than the fish??

Answer：

The original sentence was deleted. Replace with the following sentence.

These two gyrodactylidae parasites were collected from the fish fins, and these fish were harvested in China's Qinling Mountains.

Question：

Add space

Answer：

Added space in numbers betweeen unit symbols. “10 cm”

Introduction

Question：

Significant “bit” specificity, microhabitat?

Answer：

Revised this word as “microhabitat”.

Question：

Rephrase? The cited styudy shows that parasites occur more frequently/abundantly in those fins, but does not really address differences in susceptibility. In contrast, work by Buchmann et al., Lindenström at al, and by Rubio-Godoy et al, does address this topic by analyzing potential immune differences between fish fins/body regions.

Answer：

This sentence has been modified to increase the impact of immunity. References have been added. See references 2, 3, and 4 for details

Question：

True; but very important to also consider the well-documented effect of temperature on population dynamics of ectoparasites such as those studied here

Answer：

Water temperature does have an effect on the number of fish parasites. We have added quotations in this regard.

Question：

Better: on Salvelinus fotinalis fry

Answer：

The original sentence was deleted. Replace with the following sentence.

on Salvelinus fotinalis fry

Question：

attached

Answer：

Modified word, new one is “Parasites attached”

Question：

Add space

Answer：

Added spaces in symbol between word. The new format is “parasites (G. colemanensis)”.

Question：

Delete this sentence. If you are already denoting that is Gyrodactylus sp., of course means that the species is not described, other wise you will add the species name as you are doing with P. variegatus

Answer：

A newly discovered parasite has not been given a formal name. So, the original text was deleted.

Question：

Delete: which is located by the…, and the geographical position of the locarion. This shouldnt be in the introduction, just only in the materials and methods or results.

Answer：

Removed this geographic information. It is introduced in the methods and materials section.

Question：

Gyrodctylus sp. add sp.

Answer：

Modified latin name format. Gyrodctylus sp. 

Materials & Methods

Question：

Collection permit obtained? It was not stated in the submission format

Answer

We have obtained permission for field experiments. We recapitulate in the first paragraph of the Methods and Materials section.

Question：

microscope; please state if thus was the case , the type of microscope used and what magnification was used - Materials and methods should enable replication of the work reported and this important information is missing.

Answer

We have rewritten this part. The method of selecting parasites using a microscope is detailed. The type and magnification of the microscope are also described. GLM was used to analyze the effect of host length and water temperature on the location of parasites. In order to explore the specific effect of water temperature and length on the number of parasites on fins, a Spearman correlation analysis was performed.

Question：

it would be helpful if brief mention were made on how easy/complicated it is to differentiate between these two genera, as both are tiny organisms. This would also be relevant to be able to replicate this study.

Answer

We have rewritten this part. The method of selecting parasites using a microscope is detailed.

Question：

Density of infection is an unusual way to describe ecological parameters of infection; and insufficiently described here as the surface area of the fins is not considered to truly reflect density. 

Answer：

Our experiments used a direct parasite counting method. Change the density to the number of parasites throughout.

Question：

How many fish were collected?

Answer：

The number of captured hosts is hereby marked. Fish (Homatula variegata) were collected (n = 442)

Question：

You should put Acession numbers or After is species name the prefix Acc. Nos.

Answer：

We have revised numbers format. Voucher specimens of the parasites and host were deposited in the Fish Disease Laboratory, Shaanxi Normal University (Accession number: H. variegata: Acc.HV20160012; Gyrodactylus sp.: Acc.GS20160001 and P. variegatus: Acc.PV20160001).

Results

Question：

Before addressing density of infection on fins, it would be interesting/relevant, to describe overall infection trends; and only later describe dynamics on the fins.

Answer：

Our experiments used a direct parasite counting method. Change the density to the number of parasites throughout.

Question：

a bit confusing to present abundance, as this has not been presented in Materials and Methods and the first part of this paragraph mentions density.

Answer：

Question：

this whole section is very confusing, and should be carefully revised as basic concepts are not applied correctly: e.g., fins do not infect parasites, and parasites are not susceptible to specific microhabitats on the host

Answer：

Our experiments used a direct parasite counting method. The descriptions of density, intensity, and abundance have been omitted throughout the text, and the number of parasites has been unified. In addition, the influence of two factors (water temperature, host length) on the number of parasites on fins was analyzed using GLM.

Question：

check: these statements are contradictory

Answer：

We have rewritten this sentence.

Question：

I would move this after the next title, or combine this one with the next title. This part of the Material and methods are vry messy, please rewrite. Always write the name of the fih species, number of fish, place where they were collected, what did you find and then you stablish which statistical methods you used for your study

Answer：

We added content. The specific experimental contents are described, such as the fishing geographic information of the host, the body length of each host sample, the types of two parasites, and the statistical analysis method. See the first paragraph in the results section for details.

Question：

Between the Mean and SE must and space before the simbol, please add it and make it in the whole document.

Answer：

We adjusted all the formats.

Question：

Delete Figure 2A and move it after anal fins was relatively small (Figure 2A).

Answer：

We adjusted the position of the image name after this sentence. The details are as follows

The density of the infection of parasites detected on the dorsal fins and anal fins was relatively small (Figure 2A). 

Question：

Put it at the end of the sentence. The number of paraaites on different fins decreased significantly during April and May. In June, the mumber of Gyrodactylus sp. increased on the pectoral fin (Figure 2B).

Answer：

We adjusted the position of the image name after this sentence.

Question：

This should be first or put it together with the previous subject

Answer：

We merged these two paragraphs

Question：

The case is different than the rest of the MS. You can not cite first a higer Fig and then a lower one, pease change this

Answer：

This section adds GLM analysis to explore the significant effects of water temperature and host length on the number of parasites on different fins. Figure 5 in the original manuscript is similar to other diagrams, so it was deleted.

Question：

If you put this in this way, it means you did the samplin on September the following year. Otherwise you should put March-September 2016

Answer：

We have corrected the date expression. Interestingly, throughout the study period (March to September, 2016),

Discussion

Question：

Gyrodactylus sp. occurred most frequently on the pectoral and pelvic fins, while P. variegatus occurred on the pectoral fin, pelvic fin and caudal fins. contradiction

Answer：

We have modified this sentence. On the basis of HSD analysis, further analysis by GLM revealed that the two parasites were not distributed in the same position on the fins.

Question：

irrelevant infromation, not directly related to the hypotheses tested in this study

Answer：

We removed irrelevant references

Question：

The correct authority is von Nordmann, 1832, please correct.

Answer：

We have corrected the author's name. Gyrodactylus (von Nordmann, 1832)

Question：

Delete it “Fish also often appear to compete for territory [2].”

Answer：

Removed this sentence in the manuscript

Question：

This is not completely true, “It has been reported that some Monogenea parasite species have the longest life span at 21°C, and are not able to survive at 30°C [16].”

Answer：

Removed this sentence in the manuscript

Question：

References [18]

Answer：

This reference has been deleted

Conclusions

Question：

I think is rise??? please change

Answer：

Deleted rose and changed it to rise. The number of the two parasites on the fins of the host rise again in the autumn.

References

We have modified the reference format of the manuscript with reference to the Plos ONE format. Examined and corrected species names, author names in references.

Table

Due to the similar content to Table 1 and 3, we deleted Table 2.

Added Table 4, corresponding to GLM analysis.

Added Table 5, corresponding to Spearman analysis.

Figure

We changed the name of the Y axis in Figure 2.

Figure 4 was deleted because Spearman analysis was used, and the new content is in Table 5.

Deleted Figure 5, because its content is similar to Table 3.

Thanks again for your review

Sincerely

All authors

---

## [Editor Report · Decision Letter 1]

27 Feb 2020

A survey of gyrodactylid parasites on the fins of Homatula variegata in central China

PONE-D-19-26341R1

Dear Dr. You,

We are pleased to inform you that your manuscript has been judged scientifically suitable for publication and will be formally accepted for publication once it complies with all outstanding technical requirements.

With kind regards,

Ulrike Gertrud Munderloh, Ph.D.

Academic Editor

PLOS ONE
---

## [Editor Report · Acceptance letter]

3 Mar 2020

PONE-D-19-26341R1 

A survey of gyrodactylid parasites on the fins of *Homatula variegata* in central China 

Dear Dr. You:

I am pleased to inform you that your manuscript has been deemed suitable for publication in PLOS ONE. Congratulations! Your manuscript is now with our production department. 

With kind regards,

on behalf of

Dr. Ulrike Gertrud Munderloh 

Academic Editor

PLOS ONE